# Visual Pre-training for Navigation: What Can We Learn from Noise?

**Yanwei Wang**
MIT CSAIL
yanwei@mit.edu

**Ching-Yun Ko**
MIT RLE
cyko@mit.edu

**Pulkit Agrawal**
MIT CSAIL
pulkitag@mit.edu

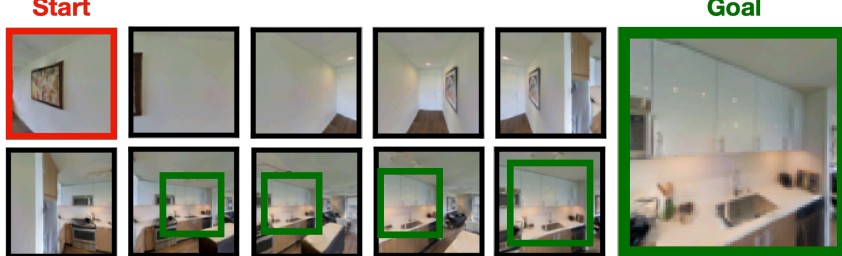

Figure 1: A rollout of our navigation policy pre-trained with random crop prediction on *synthetic noise images*. Given a goal image and the current view, the agent first explore the environment by panning its camera. Once the goal view is detected, shown by the green bounding box, the agent moves towards the goal to complete the task. **Click the image for a video.**

## Abstract

In visual navigation, one powerful paradigm is to predict actions from observations directly. Training such an end-to-end system allows representations that are useful for downstream tasks to emerge automatically. However, the lack of inductive bias makes this system data-hungry. We hypothesize a sufficient representation of the current view and the goal view for a navigation policy can be learned by predicting the location and size of a crop of the current view that corresponds to the goal. We further show that training such random crop prediction in a self-supervised fashion purely on *synthetic noise images* transfers well to natural home images. The learned representation can then be bootstrapped to learn a navigation policy efficiently with little interaction data. **Code is available at** https://github.com/yanweiw/noise2ptz

## 1 Introduction

Consider a visual navigation task, where an agent needs to navigate to a target described by a goal image. Whether the agent should move forward or turn around to explore depends on if the goal image can be found in the current view. In other words, visual navigation requires learning a spatial representation of where the goal is relative to the current pose. While such a spatial representation can naturally emerge from end-to-end training of action predictions given current views and goal images, this approach can require a non-trivial amount of interaction data as self-supervision, which is costly to acquire. In order to learn a navigation policy from minimal navigation data, we ask if we can learn purely from synthetic noise data a spatial representation that can both transfer well to photo-realistic environments and be sufficient for downstream self-supervised policy training.

NeurIPS 2022 Workshop on Synthetic Data for Empowering ML Research.

One useful spatial representation can be the relative $2D$ transformation between two views. Specifically, if we can locate a goal image as a crop in the center of the current view, an agent should move forward to get closer. If the crop is on the left/right side of the current view, the agent should turn left/right to center the heading. The relative $2D$ transformation can thus be parametrized by the location and the scale of one crop inside the current view that corresponds to the goal image, which together we refer to as PTZ factors (analogous to cameras' pan, tilt, and zoom.) Given a fixed pre-trained PTZ encoder that extracts spatial representation from pixel inputs into a low-dimensional PTZ vector, the downstream navigation policy can learn to predict actions from PTZ vectors with far fewer interaction data than learning directly from pixel inputs. Additionally, an embedding space learned directly from pixels is likely to suffer from the distribution shift as we move from training images to testing images. On the contrary, a policy that inputs the PTZ parametrization, which only captures relative transformation, will be insulated from domain shifts. The goal of this paper is to verify that bootstrapping a pre-trained PTZ predictor allows learning a navigation policy with only a little interaction data in new environments. **Our major contribution shows self-supervising the PTZ encoder to predict random crops of *synthetic noise images* produces a sufficiently performing spatial representation for navigation that also transfers well to photo-realistic environments.**

## 2    Related Works

Self-supervised learning has shown steady progress in closing the performance gap between supervised models and unsupervised models in computer vision [5, 8, 10, 21, 6, 4, 3]. Early works [5, 8, 10, 21] leverage various pretext tasks to learn transferable representations, while more recent works [6, 4, 3] focus on contrastive learning to extract useful features from unlabeled images. Similarly in robotics, [14, 18] pursue contrastive learning and data augmentation to learn image-based control policies for simulated agents, and [12, 1] use real robots to experiment with 50k grasps for 700 hours and 100k pokes for 400 hours respectively. All these works train and test on data collected in the same domain. While generating enough data to train a self-supervised robot policy in the simulation is fast, collecting a large enough real-world interactive dataset can take hundreds of hours even if the data collection is automatic without any manual annotation. An attractive idea to further reduce the need for interaction data is to only test a learned policy in the real-world distribution and use alternative data sources for training. For example, [19] discovers visual pre-training on object detection significantly improves sample efficiency for learning affordance prediction; [20] investigates representation learning on a single image; and [7, 9] show synthetic data can be a potential cheap replacement for real-world data. Specifically, [7] generates a synthetic fractal noise dataset FractalDB for training, and [9] demonstrates the utility in pre-training Vision Transformers (ViTs) with FractalDB. Beyond fractal noise, [2] provides a comprehensive study on how different noise types affect representation learning.

## 3    Method

Given a robot interaction dataset $D$, we want an agent to navigate to a goal location upon receiving a goal image $x_g$. The dataset $D$ consists of image action pairs $(x_t, a_t, x_{t+1})$, where $x_t$ is the pixel observation at time $t$, $a_t$ the action taken at time $t$, and $x_{t+1}$ the pixel observation after the action. One self-supervised approach to learn a navigation policy is to train an inverse model on $D$, which predicts an action given a current view and a goal view [1]. We divide the inverse model into an encoder $E_\phi$ that encode image pairs into states and an LSTM policy $\pi_\theta$ that predicts actions given states as $\hat{s}_t = E_\phi(x_t, x_{t+1})$, $\hat{a}_t = \pi_\theta(\hat{s}_t)$. We can train $E_\phi$ and $\pi_\theta$ jointly end-to-end by minimizing cross entropy loss between $a_t$ and $\hat{a}_t$ for all image-action sequences in $D$ as shown in the top part of Fig 2 (a). We use a simple LSTM architecture to learn a policy as memory of past states and actions benefits navigation with only partial observations [11], and we discuss in the experiments section that the LSTM is sufficiently expressive to solve our navigation task.

To improve data efficiency, we observe interaction data (i.e. action labels) is only necessary for training policy $\pi_\theta$. To find alternative cheap data sources for training $E_\phi$, we observe in Fig 2 (c) that a goal view can be seen as a crop from the current view. The relative location and size of the crop indicate the relative heading and distance of the agent from the goal location. Thus the spatial relation between the two views can be parametrized by the panning angle $p$, tilting angle $t$, and zooming

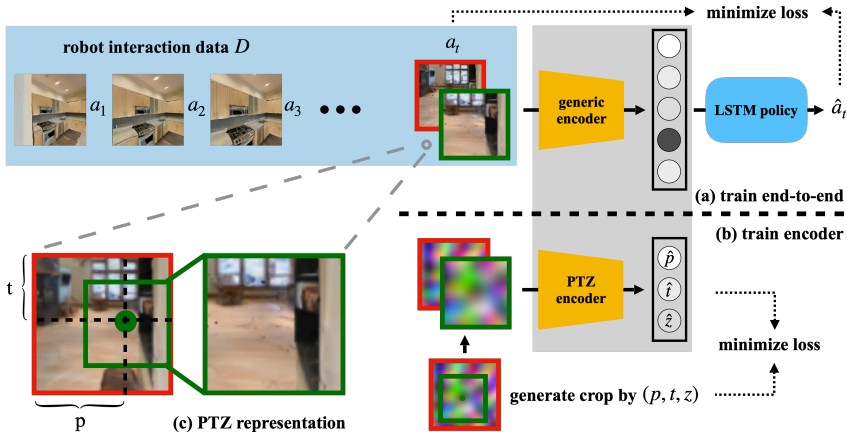

Figure 2: Top ($a$): An end-to-end system, where a generic encoder is jointly learned with a LSTM navigation policy. Bottom: (b) Replacing the generic encoder with a fixed PTZ encoder pre-trained from synthetic noise images reduces the amount of interaction data needed to train the navigation policy.

factor $z$ of a camera. We hypothesize such a 3 DOF parametrization is sufficient for local navigation where a goal is close to the agent. The low dimensionality of this state representation $\hat{s}_t$ is desirable as the mapping between states and actions can now be learned with a little interaction data. We will now discuss our implementation of the PTZ encoder that predicts $(p, t, z)$ given two images.

### 3.1 Self-Supervised Training of PTZ Encoder

We approximate learning a PTZ encoder with learning a random crop predictor. Given a $256 \times 256$ image, we randomly crop a $128 \times 128$ pixel patch to form the current view. For the goal view, we randomly sample a scale factor $z$ from $0.5 - 1$ and $(p, t)$ from $0 - 1$ relative to the top left corner of the first crop to generate the second crop at the location $(128x, 128y)$. We resized the second crop to a $128 \times 128$ pixel patch afterward. Additionally, we generate pairs of crops without any overlap to supervise scenarios where the goal image is not observable in the current view. We assign PTZ label $(0, 0, 0)$ to indicate zero overlap. We train a ResNet18 network to regress the PTZ factors $(p, t, z)$ on concatenated crop pairs from static natural home images (without action labels) or synthetic images as shown in Fig 3.

### 3.2 Self-Supervised Training of PTZ-enabled Navigation Policy

Given an interaction dataset $D$ collected from random exploration, we encode the current and goal views into states using the PTZ module as a fixed feature extractor and train an LSTM policy $\pi_\theta$ to predict actions from the states. To train $\pi_\theta$, we sample sub-trajectories up to a maximum trajectory length from every image-action sequence in $D$. We use a single-layer LSTM network with 512 hidden units with ReLU activation to regress the navigation action with L1 loss. During inference, the LSTM predicts an action autoregressively until the agent reaches the goal or the episode terminates. Notice if sub-trajectories of the maximum sequence 1 are sampled, the LSTM policy is effectively trained as a feed-forward network that does not use memory from previous time steps.

## 4 Data Collection

### 4.1 Interaction Data Collection

For training, we choose ten Gibson environments—'Crandon,' 'Delton,' 'Goffs,' 'Oyens,' 'Placida,' 'Roane,' 'Springhill,' 'Sumas,' 'Superior,' and 'Woonsocket.' We create a 20k/10k training/validation set and a 50k/10k training/validation set by sampling 40/20 and 100/20 starting locations in each of the ten environments. We also create a small 1k/1k training/validation set by sampling 20 starting locations from 'Superior' and 20 starting locations from 'Crandon' respectively. Collectively, we

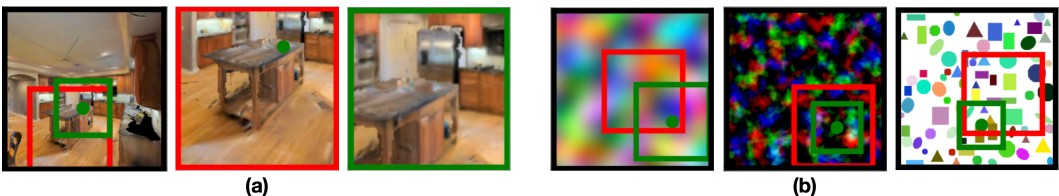

Figure 3: Sampling random crops of (a) static kitchen scenes and (b) synthetic noise images (From left to right: Perlin noise, fractal noise, and random geometric shapes.) Red boxes denote the current view, while the green boxes denote the goal view.

have created three interaction dataset $D_{2k}$, $D_{30k}$ and $D_{60k}$. For our random exploration strategy, we refer the readers to [11].

## 4.2 PTZ training Data Collection

First, we generate a training set from similar domains to the navigation experiments. Specifically, we sample 6500 photo-realistic home images sourced from 65 Gibson environments rendered by the Habitat-Sim simulator to form the training set and 2300 home images from 23 other Gibson environments to form the test set. Notice these images are generated i.i.d. without any action labels. We refer to this dataset as $D_{habitat}$

Second, we generate Perlin noise and fractal noise using [16]. Perlin noise is generated from 2, 4, 8 periods and fractal noise is generated from 2, 4, 8 periods and 1-5 octaves. We generate 10k Perlin noise, 10k fractal noise, and 20k random shapes to form a 40k noise dataset $D_{all\_noise}$. However, this particular composition of noise is rather wishful as we do not know yet which one is the best for PTZ encoder training. To uncover which noise is the best surrogate data source for natural home images, we also create a 40k $D_{perlin}$, $D_{fractal}$ and $D_{shape}$, each containing only one kind of noise.

## 4.3 Noise Choice for PTZ Pre-training

We first tried training our PTZ encoder on Gaussian noise. The resulting poor performance suggests the particular choice of noise is critical. We hypothesize that patterned noise rather than high-frequency noise should be more useful as the encoder probably needs some visual cues to find relative transformations. To this end, we include Perlin noise, which can be used to simulate cloud formations in the sky, and fractal noise, which can be found in nature [7], in the dataset to train the encoder. We further include random geometric shapes as they are found in man-made environments and can help the encoder learn edges and orientations. A sample of these three different kinds of random noise is shown in Fig 3. We follow the same procedure as before to sample random crops on these noise images. Using noise for pre-training completely removes the need to access a testing environment for training data.

# 5 Experiments

## 5.1 PTZ Encoder Evaluation

To verify the PTZ encoder's utility in visual navigation, we test the pre-trained encoder on 2300 natural home images. Specifically, we evaluate the model by calculating the IOU between the ground truth bounding box and the predicted bounding box when the goal can be at least partially seen in the current view. If the two images are not overlapping, we calculate the success rate at which the model predicts a pan and tilt pixel center that is within 10 pixels away from the ground truth label $(0,0)$. This corresponds to no detection of the goal in the current view. To test if the PTZ encoder trained on synthetic noise images can perform well when evaluated on natural home images, we generate random crops from five different data sources: $D_{habitat}$ (Gibson environments loaded in Habitat simulator [13]), $D_{all\_noise}$ (all three synthetic noise combined), $D_{perlin}$ (Perlin noise only), $D_{fractal}$ (fractal noise only) and $D_{shape}$ (random shapes only). While the concurrent training of a PTZ encoder to predict non-overlapping and overlapping crops of $D_{habitat}$ gives near-perfect evaluation results, the simultaneous training of the two prediction tasks on noise images proves slow to convergence. We

| Data | Overlap-IOU | Non-Overlap |
|------|-------------|-------------|
| Shape | $72.1 \pm 0.4\%$ | $48.2 \pm 1.1\%$ |
| Perlin | $61.6 \pm 0.4\%$ | $65.3 \pm 0.6\%$ |
| Fractal | $87.3 \pm 0.5\%$ | $80.1 \pm 0.6\%$ |
| All noise combined | $92.2 \pm 0.1\%$ | $93.2 \pm 0.5\%$ |
| Habitat | $97.1 \pm 0.1\%$ | $98.8 \pm 0.1\%$ |
| Habitat w/o non-overlap | $96.4 \pm 0.1\%$ | $1.5 \pm 0.1\%$ |
| Fractal w/o curriculum | $78.0 \pm 0.4\%$ | $2.7 \pm 0.3\%$ |

Table 1: Performance comparison of PTZ encoders on the 2300 natural home image test set. In the case when the given goal view (partially) overlaps with the current view, we use the IOU between the ground truth box and the predicted bounding box of the goal image in the current view as the evaluation metric. In the case when the given goal view does not overlap with the current view, we set the ground truth PTZ label to (0,0,0). The corresponding success is defined by whether the encoder predicts a $(p, t)$ that is close enough to (0,0). We report the success rates in such non-overlap cases.

hypothesize that noise images lack salient structures for the PTZ encoder to easily establish spatial relationships between two overlapping crops, and consequently training with non-overlapping crops concurrently could complicate the training. Therefore, we first train the encoder to only predict PTZ for overlapping crops until convergence before mixing in non-overlapping crops, which results in high prediction accuracy for both non-overlapping and overlapping crops. We call this staggered training a curriculum for PTZ training with synthetic noise. Once we have the pre-trained PTZ encoder $E_\phi$, we fix its weights and optimize only the LSTM weights in $\pi_\theta$ as we train the navigation policy with interaction data $D$.

In Tab 1, we show the mean and standard deviation of inference performance on both overlapping and non-overlapping image pairs of PTZ trained on different data sources. Training on natural home images $D_{habitat}$ naturally produces the highest accuracy. However, we observe that training on all three noises combined $D_{all\_noise}$ produces competitive results without seeing a single natural home image. **This suggests that PTZ of two views is independent of the underlying visual statistics and can transfer well from one domain to another.** This property allows for stable training of downstream LSTM policy as PTZ representation will be consistent across different visual domains. This also suggests we do not need to collect new data to fine-tune our navigation policy if we move from one environment to another. We show qualitative inference results of the PTZ encoder in Fig 1 where the green bounding boxes indicate where the PTZ encoder predicts the goal crop in the current view. To understand which noise is the most helpful for pre-training, we train the PTZ encoder on individual noise $D_{perlin}$, $D_{fractal}$ and $D_{shape}$. We see in Tab 1 training on fractal noise to convergence outperforms Perlin noise and random shapes and approaches the performance of all noise combined. This result is in line with the finding in [7] and indicates that the natural home images may share more similar visual statistics with fractal noise than others.

## 5.2 Navigation Policy Evaluation

To test our hypothesis that a pre-trained PTZ encoder can improve data efficiency, we consider a local navigation task, where the goal is in the same room as the agent. We choose five Gibson environments [17]—'Beach,' 'Eastville,' 'Hambleton,' 'Hometown,' and 'Pettigrew.' and evaluate our PTZ-enabled navigation policy on a multi-step navigation task. Specifically, we sample 30 starting and goal locations in each of the testing environments such that the start and the goal are five forward steps apart. We then randomize the heading such that the goal can be in any direction including behind the agent. To infer a trajectory, the agent will auto-regressively predict the next action given the current view and goal view until it uses up to 50 steps or reaches the goal. To determine if an agent arrives at an observation that is close enough to the goal image, we use perceptual loss [22] to measure the similarity between those two observations in the eye of a human. If the similarity score exceeds a threshold of 0.6 while the agent is within a 0.5m radius of the goal location, we consider that agent has successfully reached the target.

We see in Fig 4 (a) that without PTZ encoding, the success rate of navigating to the goal location increases as we train the whole pipeline with more data and longer trajectory sequences, presumably because the system has to learn the appropriate state representation from scratch. However, with PTZ encoding (trained with all noise) such dependency on interaction data becomes less acute.

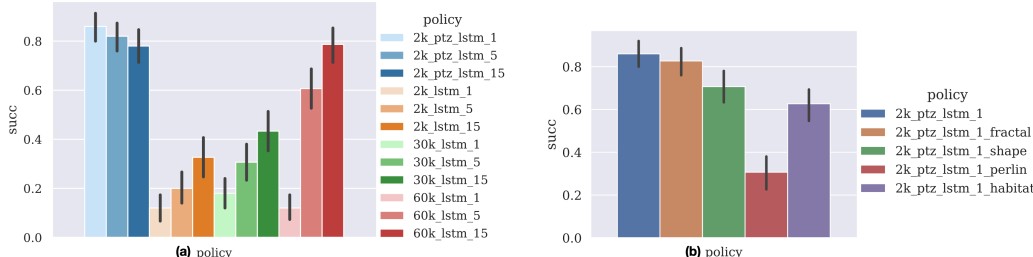

Figure 4: (a) PTZ-enabled (all noise) navigation policy (blue) trained with 2k interaction data outperforms the best end-to-end policy (red) trained with 60k interaction data. Prefix 2k, 30k, 60k denote the size of data, while suffix 1, 5, 15 denote the maximum length of sub-trajectories used for training. (b) An ablation study showing a PTZ encoder trained on fractal noise alone (orange) is almost as good as one trained on all noise combined (blue) in enabling the downstream navigation policy.

Specifically, training a PTZ-enabled policy with only 2k interaction data and one-action sequences already outperforms training an end-to-end system from scratch with 60k data. As we increase the action sequence length to train the PTZ-enabled system in the low data regime (2k), the performance actually drops. Note training an LSTM with a single action step is essentially treating the LSTM as a feed-forward network. The fact that a PTZ-enabled feed-forward policy outperforms PTZ-enabled LSTM policies, which in this case likely overfit to the small interaction dataset, suggests we do not need to consider more expressive architectures such as Transformer [15] to learn a PTZ-enabled policy.

To further investigate which noise type helps train the PTZ module the most, we show in Fig 4 (b) that the PTZ encoder trained with fractal noise outperforms Perlin noise and random shapes. Although the evaluation metrics in Tab 1 shows a PTZ encoder trained with fractal noise alone is less accurate than one trained with all noise combined, the navigation results show that it is still sufficient to achieve a high success rate for the downstream navigation task using a PTZ encoder trained with fractal noise alone. Lastly, we show in Fig 4 (b) that a PTZ encoder trained on natural home images yet only with overlapping crops ('2k_ptz_lstm_1_habitat') leads to poorer navigation results than a PTZ encoder trained on synthetic noise but with both overlapping and non-overlapping crops ('2k_ptz_lstm_1'). Consequently, it is essential to train a PTZ encoder to recognize when two views are overlapping through a curriculum of training first on overlapping crops followed by adding non-overlapping crops.

# 6    Conclusion

In this paper, we focus on visual pre-training for navigation. As training in an end-to-end fashion requires a significant amount of data (60k as shown in Figure 4), we break the system into two modules: a feature encoder module (PTZ module) and an LSTM policy module, where the first part can be effectively pre-trained without the use of expensive interaction data. Three synthetic noise are included in pre-training the PTZ module and their effectiveness are extensively evaluated. Promising experimental results verify the usefulness of a PTZ encoder in reducing the need for interaction data.

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
