# OpenReview forum: "Visual Pre-training for Navigation: What Can We Learn from Noise?"
_NeurIPS.cc/2022/Workshop/SyntheticData4ML — Neurips 2022 SyntheticData4ML_

### Official Review · Reviewer_xvEo · 2022-10-16
**Great idea**

**Rating:** 7
**Confidence:** 3

**Review:**

# Summary
The paper proposes to pre-train a navigation policy with a novel self-supervised objective based on random crop predictions using synthetic noise images. This pre-training technique can help to reduce the need for interaction data, which is rare in real-world settings

# Strengths
* the problem is well-motivated and applicable to many robot interaction settings
* the method is simple and easy to implement

# Suggestions
* it remains unclear why the random crop predictions, as shown in Figure 3(b), actually help to learn PTZs
  * the ablations provided in Table 1 are good, but it would be nice to see more analyzes on whether we actually need three different noise sources or just more data of one
* the idea is similar to SSL in RL, e.g., CURL or DrQv2; the connection could be discussed
* "We observe that training non-overlapping crops and overlapping crops concurrently can lead to conflicting gradients." What does that mean? how is it observed? please clarify in the next version

---

### Official Review · Reviewer_nrE7 · 2022-10-16
**Well-written paper where noise is successfully used to train a navigation policy model**

**Rating:** 7
**Confidence:** 4

**Review:**

In this paper, the authors use different types of noise to train both a PTZ (pan, tilt, zoom) encoder and a PTZ-based navigation policy LSTM network. It is shown that using only noise-based pre-training leads to strong performance.

Pros:
- The motivation and methodology are both well-written and easy to understand.
- The experimental evaluation shows that using synthetic (noise) data can achieve strong performance for both a PTZ encoder and a learned navigation policy network (an LSTM). Although the PTZ encoder trained on noise performs worse than that trained on real data (which is to be expected), their experiments show that it is able to transfer the knowledge learned from noise to real-world data.


Cons:
- It is not clear why the network trained with real data (Fig. 4b, 2k_ptz_lstm_1_habitat), even though it leads to best encoder performance (Tab. 1). Is the navigation policy trained with noise, even though the encoder is trained with real data? More details on this experiment should be given.
- It is difficult to know how this encoder-based pre-training would transfer to other visual tasks. This, however, is a minor issue because this paper mainly deals with the navigation task.

---

### Official Review · Reviewer_GTkG · 2022-10-18
**Nice contribution to visual navigation**

**Rating:** 7
**Confidence:** 4

**Review:**

This paper proposes to train one aspect (the image encoder) of a visual navigation system using artificial data.  The system learns to move the modeled camera to zoom in on  a given crop from the image.
PROS:
The method is simple but works well.
The curriculum experiments are insightful
Many ablation experiments offer more insight
Demo is nice

CONS:
The description of the experiments in Fig 4 could be clearer

---

### Meta-Review · Area_Chair_LMZk · 2022-10-19

**Recommendation:** Accept